# Public and Private Standards in Crop Production: Their Role in Ensuring Safety and Sustainability

**Salvatore Squatrito, Elena Arena, Rosa Palmeri and Biagio Fallico \*** 

Dipartimento di Agricoltura, Alimentazione e Ambiente (Di3A), University of Catania, Via S. Sofia 98-100, 95123 Catania, Italy; salvatoresquatrito@gmail.com (S.S.); earena@unict.it (E.A.); rpalmeri@unict.it (R.P.)

**\*** Correspondence: bfallico@unict.it; Tel.: +39-0957580214

**Abstract:** From the comparison of regulations and/or standards for the organic, conventional and/or integrated citrus production method and a voluntary certification, it emerges that farms certified with voluntary non-regulated certification systems, such as the IFA FV GLOBALG.A.P, are obliged to take into account the highest number of aspects, reported in a more complete register, than the organic ones. Moreover, this is also supported by a continuous-time planned process of revision and updating of the applicable versions of the standard. The environmental impact of the food production, the safety aspects of food products, as well as the health, ethics, and safety aspects of workers, are largely considered and inspected in the GLOBALG.A.P., while the organic system, despite the IFOAM suggestions and indications, is only considered partially. This means that, from a practical point of view, the organic product can be considered "clean and safe", but not more environmentally friendly than the GLOBALG.A.P. products.

**Keywords:** organic; oranges; citrus; GLOBALGAP; audit; certification; food control

## 1. Introduction

All agricultural holdings, nowadays, among the numerous obligations, must take care of, whatever method of production they adopt, any crop operation they perform, any type of crop protection product they use. They are obliged to keep a register of treatments (or orchard register) where they promptly take note of all the cultivation activities carried out, the products purchased and used on the crops. This is a cogent fulfillment that becomes very important when it refers to the treatments carried out, with the specification of the used plant protection products, the amount, and the date. It determines and constrains harvesting, guarantying food safety, and traceability of products. This is, therefore, aimed at obtaining productions that are increasingly suited to the demands and needs of the final consumer, who, being increasingly aware, requires guarantees both on the quality of production, safety and that it is eco-conscious and healthy. The agri-food system is under a transition to be sustainable.

The necessity of explicit use of social-ecological systems concepts, [1] as well as the use of assessment tools [2] in food sustainability systems, has been recently highlighted. The increasing importance of sustainability assessment in food processing can be seen in recent studies aimed at assessing sustainability, in terms of environmental end energy performance, of food processing facilities [3,4]. Moreover, due to the growing number of people living in metropolitan areas, a holistic sustainability assessment method for the urban system has been developed [5]. To this and other purposes, FAO published very useful guidelines for the Sustainability Assessment of Food and Agriculture System (SAFA) [6].

The idea of sustainability is still ubiquitous depending on multiple, interdependent facets or 'pillars'. Among many suggested formulations, a three-aspect model seems to prevail: The environmental, the economic, and the social aspects. Most of the stakeholders in the food chain, although they agree

with the definition, argue that social sustainability or responsibility actions can be justified on moral grounds as 'the right thing to do'. However, they usually have to be backed up in more pragmatic terms as a way of 'doing well by doing good' [7]. Other studies [8] investigating the role of food retailers in the transition towards more sustainable agri-food systems, affirmed that systemic ethics and inclusive governance of the food supply chain are key features for initiatives to contribute to a sustainability transition.

Moreover, to keep in account the above reasons, the methods based on good agricultural practices (GAP), integrated production (IP), and organic agriculture (OA), on which the requirements of private voluntary standards are able to guarantee an accurate control over the processes, are increasingly exalted.

Data concerning the certified organic food products in Italy in 2017 return a scenario of absolute growth over the previous year, both in terms of cultivated areas, which amounted to over 1.9 million hectares (+6.3%), number of controlled operators, now around 76,000 (+5.2%), and consumption (+16.6.2%) [9]. The perception and the awareness of the EU on food safety [10], the interaction between conventional and organic agriculture, as well as their impact on the value of agricultural, have been reported [11]. The usefulness of a tri-partite system (standard-setting, certification, and accreditation) in the food organic agriculture has been pointed out [12]. Moreover, the studies carried out both on safety [13–15] and quality [16] aspects in some of the most typical Italian food chains bear the significance of these issues. The differences in the nutritional aspects of organic and conventional food products have been also investigated [17].

Regarding the private certification schemes, whose control has been entrusted to independent third-party organizations, it is clear that their rapid success at the global level has allowed the operators present on the market and the companies of the same large-scale distribution to transmit information to the final consumer on the level of food quality and safety achieved by the agri-food supply chains and by production. On the other hand, the retailers' power, due to their position at the head of the food supply chain, is increasingly raising questions of legitimacy in imposing private food standards [18], the risk of "ritualism" around the audit activity has also been pointed out [19].

Over the years, the need to have more in-depth information on food products has been strengthened, to have useful information to confirm the consumption of healthy and safe products with an "identity card", able to trace all the cultural and manipulation interventions that they have brought "from the field to the fork". What has been said above has become even more important because the farm has in many cases lost its function solely of production, to evolve and take on different connotations with the offer of other services (multifunctional company), such as that of direct sales point "from producer to consumer", farmhouse, educational farm where external users are in direct contact with the crops and the fruits derived from it. This requires total compliance with Reg. EC no. 178/2002 [20], which requires that food business operators must have systems and procedures that allow traceability of food products and products used for the production of food [21]. To this purpose, the orchard register takes on a very important and even more indispensable role.

The aims of the present study are, therefore, to: Provide an overview of the binding regulation of the sector and the operating procedures concerning the registration of the cultivation activities of primary products by the agricultural operators, to verify the different annotation modalities envisaged between the organic and the conventional production methods with reference to the obligations imposed by the respective regulations, and then to compare them with the corresponding registrations foreseen by private certification schemes, such as the IFA FV GLOBALG.A.P. Moreover, the control and supervisory systems operating by public or private authorities and the sanctioning systems were highlighted.

## 2. Materials and Methods

To better understand the workings of the public authority inspection process of organic products (OP) in Italy, the authors conducted expert interviews with senior government officers in charge of the program, namely ICQRF (Ispettorato Centrale Repressione Frodi), the branch of the Italian Agricultural,

Food and Environment Ministry in charge of guaranteeing the quality of organic products [22]. They interviewed the head of the program in Catania (Italy) to determine the objectives of the program, its organization, its size, and current challenges, using the citrus production as a case study. The interviews were also aimed to better understand how the auditing is being conducted.

Using a structured questionnaire (Table 1), they further interviewed four inspectors of certification bodies and the manager of an Organic Producer Organization, representing about 50 citrus farmers. The survey involved a twenty-four-month recall period, from January 2017 to January 2019, and recorded detailed information on crop management practices, pesticide use and handling, household characteristics and, overall, the checklists and the reports of inspections.

**Table 1.** Description of the questionnaire used for interviews.

| | Questionnaire | Questions |
|---|---|---|
| **Food Companies** | Knowledge of applicable legislation in Agricultural and Food sector | • What are the most important laws for your sector?<br>• Do you know the: D.L 220/1995, Reg. CE 178/2002?<br>• Do you know DL 150/2012; Reg.s EU 834/07; 889/08; 392/2013 and the newest Reg EU 848/18?<br>• Other applicable laws? |
| | Recording | • What kind of records you have in your company (farming activities, pesticides, etc.)?<br>• What kind of support do you use (on paper, electronic, etc.)?<br>• Do you have any record concerning the use of water, fuels, electricity, others? |
| | Inspections | • Do you have an idea about who can run an inspection in your company?<br>• Could you describe the inspection of the public authority?<br>• (If applicable) Could you describe the inspection of the control body (Organic) or the certification body (GlobalGgap)?<br>• Please, describe the inspection:<br>　- was it pre-agreed or not<br>　- what did they ask (documents, products, personal, etc.)?<br>• Other? |
| | Sanctions | • Do you have an idea about the applicable sanctions?<br>• Do you know the following laws?<br>　- LD 150/2012<br>　- M.D. 15962/2013<br>　- LD 20/2018<br>　- Other? |
| **Public Authority (ICQRF)** | Applicable legislation in Agricultural and Food sector | • Please, could you describe the most important laws applicable during inspections (paying particular attention to organic production)? |
| | Inspections | • Could you describe the inspection activities (documentary or on-the-spot)?<br>• Do you use a checklist?<br>• Please, describe the main points of the checklist? |
| | Sanctions | • Could you describe the most important sanctions for:<br>　- All the food companies<br>　- Organic companies |
| **Control Body (Organic)** | Applicable legislation | • Please, describe the applicable legislation for food companies under your control, |
| | Inspections | • Do you have a scheduled control plan?<br>• Do you use a checklist?<br>• Please, describe the main points of the checklist? |
| | Sanctions | • What is/are the applicable sanctions in case of non-conformity? |
| **Certification Body (Globalgap IFA FV)** | Applicable legislation | • Please, describe the applicable legislation for food companies under your control, |
| | Inspections | • Do you have a scheduled control plan?<br>• Do you use a checklist?<br>• Please, describe the main points of the checklist? |
| | Sanctions | • What is/are the applicable sanctions in case of non-conformity? |

## 3. Results and Discussion

### 3.1. Binding Legislation—Main National Horizontal Provisions

All the food operators interviewed agreed in indicating a greater complexity of the tasks to be performed for all operators. This is often seen more as a bureaucratic burden than an effective control system. The discussion with the members of the public authority supervisory bodies highlighted as keystones the legislation listed below (Table 2).

**Table 2.** Main points of horizontal binding legislation.

| Binding Legislation | Key Points |
|---|---|
| Presidential decree N 290 of 23 April 2001 | • Professional user<br>• Handling and prescriptions for buyers and users of pesticides<br>• Register of treatments or orchard |
| Directive 2001/128/EC | • Sustainable use of pesticides<br>• Compulsory training<br>• Reduction of risk from pesticides |
| Italian Law 150 of 14 August 2012 | • Adoption of pecuniary administrative sanctions. |

In Italy, the provision that gave rise to the obligation, all the agricultural holdings, to keep the orchard register is the Presidential Decree No 290 of 23 April 2001 (concerning the authorization and placing on the market of plant protection products) [23]. It introduces the figure of the professional user as "a person who uses plant protection products during the professional activity, including operators and technicians, entrepreneurs, and self-employed workers, both in the agricultural and in other sectors". The same decree establishes the handling and the prescriptions that buyers and users of plant protection products and adjuvants of plant protection products: (a) Must properly preserve, for the period of one year, the purchase invoices of products with a very toxic, toxic and harmful hazard classification, (b) must keep at the company, by the user, who must sign it, a register of the treatments carried out, noting within thirty days of the purchase: (1) The personal data relating to the company, (2) the denomination of the treated crop and its extension expressed in hectares, as well as the dates of sowing, transplanting, start of flowering and harvesting, (3) the date of treatment, the product, and the relative quantity used, expressed in kilograms or liters, as well as the adversity that made the treatment necessary. The following year the Ministry of Agriculture and Forestry Ministry published the Circular 30 October 2002 [21], confirming the provisions of Presidential Decree No 290 also for Autonomous Regions and Provinces.

Subsequently, the EU legislation was further developed with the Directive 2009/128/EC of 21 October 2009 [24] which establishes a framework for community action for the sustainable use of pesticides by introducing, de facto, compulsory training for users, consultants, and distributors of plant protection products for professional use, attention to the proper maintenance of the equipment used for the application of the same and national action plans to define the objectives and identify the measures for the reduction of impact and risks to human health and the environment resulting from the use of plant protection products for integrated crop protection. Italy transposed the Directive by Law no. 150 of 14 August 2012 [25], with the introduction of pecuniary administrative sanctions for the lack of periodic functional control of the equipment used for the treatments and the non-maintenance of company registers. This decree finally defines, clearly and unequivocally, the orchard register or the register of treatments and its use as "a company form that shows chronologically the list of treatments performed on different crops, or alternatively, a series of distinct modules, each related to a single agricultural culture".

Therefore, it does not provide a specific form of the orchard register, which can be freely set and proposed by the user, as well as a simple module where it is compulsory to schedule the annotation

of the treatments carried out with all the plant protection products used in the company, classified as very toxic, toxic, harmful, irritant, or unclassified. The register must report the distribution of the crop, the date of treatment, the culture and the treated area, the product used, the total amount used, the adversity in the orchard and the signature of the person who used the product. For companies that adopt agricultural production systems that refer to integrated agriculture, the record of fertilization and irrigation inputs must be registered.

The above requirements clearly show that all the EU agricultural holdings, in the present case the Italian ones, independently of a specific production scheme (organic, integrated, etc.), are obliged to follow the principles of integrate agriculture and to give, at any time, as preventive action, the pieces of evidence and the records of their farming activity. All these requirements are aimed at the highest consumers' guarantees. In fact, the safety of EU food products is not based only on the final control of products (e.g., pesticide analyses of samples), but, overall, on the ability to control the entire food supply chain [22]. This is not risk-free, particularly for farmers, in terms of complicating elements, costs, and market opportunity. In fact, most of the people surveyed complain of the system that does not take into account the dimension of the company. Therefore, for small ones, these could represent high bureaucracy and costs. Moreover, quite often, they are not able to communicate these differences to the final consumers. There is no possibility to differentiate themselves from those systems based only on the final control of food products.

### 3.2. Voluntary Legislation—EU Regulations in Organic Farming

In Europe, countries such as France and the United Kingdom, in the past 50 years, have been playing a key role in developing organic farming. The first EU regulation about this was developed in 1991 [26] and came into force the following year. In 1999, additional rules for production, labeling, and inspection of the main animal species were also developed [27]. According to these regulations, only products that have been produced and processed following the EU regulation on organics can be marketed in the EU as "organic". Moreover, all the EU member countries were asked to develop national legislation to allow the implementation of these regulations. In the early 2000s, the European Union also introduced the "Organic Farming-EC Control System" label to be used, on a voluntary basis, by producers whose products are in compliance with the EU organic regulation. Every country in the EU was asked to appoint a "reference authority" who is ultimately responsible for making sure that EU organics rules are followed. In Italy, this is run by the Ministry of Agriculture, through one of its departments, namely Ispettorato Centrale della Tutela della Qualità e Repressione Frodi dei Prodotti Agroalimentari (ICQRF) [22]. This authority can delegate its role to other (both private and public) control authorities. Once a year, EU countries report to the European Commission on the results of the controls carried out on organic operators and the measures taken in case of non-compliance. In practice, although with different aims and duties, this means a two-tier control system in organic production: The Public Authority and the Certification Body. However, often, from the farmers'/companies' point of view, the differences between the aims of the two inspections are not always such clear. Here, follows the description of the main legislation and the inspection/audit system of the public authority and certification body.

EU legislation, on organic food products, has been brought up-to-date with the: EC Reg. 834/2007 of 28 June 2007 (repealing EEC Regulation No. 2092/91, [26]) [28], EC Reg. 889/2008 of 5 September 2008 (on the application of EC Regulation No. 834/2007 regarding organic production, labeling, and controls) [29], EU Reg. 392/2013 of 29 April 2013 [30] (on organic production and amendments to EC Reg. 889/2008) that better defines and clarifies the management of the operators' control system, the plan and the operator control file and the supervision by the competent authorities of the control bodies themselves, up to the EU Reg. 848/2018 of 30 May 2018 [31] (on organic production and repealing the EC Regulation n. 834/2007). Unfortunately, the test of the last regulation [31] received strong opposition of all the Italian MEPs who voted against it, because the text, among other innovations, was going to consider less restrictive threshold levels in case of accidental contamination of not allowed

residues of crop protection products. According to their opinion, when the difference between organic and conventional food products is based only on the outcome of pesticides analysis and using less restrictive thresholds, there is a high risk for the consumer being confused and missing the real difference between organic and conventional. However, authors believe that this fear makes sense if, in future: The audits/inspections, instead of verifying all the involved processes, will be restricted to the results of analyses. The very idea of the organic product will be limited to the absence of pesticides. In the first case, it would deny one of the EU food control system pillars: The process control approach [20,31]. In fact, the control of the final product has to be considered a second level control. It cannot substitute the control of the process. Strictly speaking, it should confirm the efficacy of control of the involved processes. In the second case, the suggested principles of organic farming are questioned [32]. As a result, this means that Italy will be allowed to continue to not consider as organic product samples containing traces of pesticides, as concerns the internal products. However, at the same time, it has to guarantee the free circulation of organic products from abroad which also come with different limits. The EU Reg. 848/2018 [31], has been waiting for technical attachments and, after final approval by the Council, will enter into force on 1 January 2021.

The effective enforcement of EU standards regarding organic farming by the Member States depends also on the national specific provisions on the subject (Table 3). However, it can be noted that, despite the review of the EU legislation in Italy, the provision of implementation of the aforementioned EU legislation, until recently, was constituted by the Legislative Decree No 220 [33], 17 March 1995, (on the implementation of Articles 8 and 9 of EEC Regulation No 2092/91) [26], which requires all the operators to be included in a national register and prescribes that the business operators must keep record of all activities, according to models reported in the Annex V (points 1 to 5) of the decree. The company register proposed by the aforementioned Decree No 220 [33] was on paper, divided into four sections (raw material, crop, product preparation, and sales). The compilation of the crop sheet required records about: The type of operation and the date, the species and variety on which the intervention is performed, the number of plot and the surface, the raw material used (e.g., fertilizers or plant protection products), and the amount. The decree was not amended or updated for a long time. However, over the years, due to some misapplications, both by the public authority and the private certification bodies of organic products, it has become clear that some aspects of this decree needed to be updated and amended. In fact, in the meantime, some of the certification bodies updated the company record form by inserting additional missing information. Further, they proposed their own models in order to oblige operators to integrate the information necessary to ensure the traceability of their productions and then replace or complement them with the old register. The new one was preapproved by ACCREDIA—the Italian Accreditation Body [34]. It seems that a double system has been in force for a few years: One based on the official decree (no 220, 1995), the other one updated by the certification bodies and approved by ACCREDIA. Among the public authorities, the first who updated the Decree No 220 (1995) were some local authorities, for instance, the Autonomous Province of Trento and Bolzano (note of 09.02.2015-Prot.S164/2015), that communicated to the control bodies, operating in their area, the new model of the orchard/company registry, stating that "compared to the previous models, the current version has been revised and updated to avoid the burden of filling in more registers", a clear and unequivocal sign of the need felt by several parties to equip themselves with one tool that has been renewed and better adapted to legislative and market obligations. The Decree No 220 [33] has been replaced by the publication of Legislative Decree No 20 of 23 February 2018 [35] (related to the legislation on controls on agricultural production and organic food and agriculture). What is highlighted in the new decree is that, among the obligations of the operators, the Art. 9-point f indicates to "record all operations concerning the production and marketing of organic products, or in conversion, on special registers, or, alternatively, on mandatory registers already used in fulfillment of other regulatory provisions, provided that contain the information required by the community and national legislation for the organic sector " thus, defining how the company register should be set up,

without proposing a new model as had been done by Legislative Decree No 220 in the annexes by which it was accompanied.

**Table 3.** Legislation in organic farming.

| | Legislation | Key Points |
|---|---|---|
| EUROPEAN | EEC Regulation No. 2092/91 of 24 June 1991 | • Organic farming procedures<br>• Labeling of organic products<br>• Organic farming EC control system<br>• National reference authority<br>• Plan of controls<br>• Report to EU once a year from national authorities<br>• Delegate of national reference authority |
| | Reg. 834/2007 of 28 June 2007 | • Introduces the concept of biodiversity and rural development<br>• Organic certification of wine, aquaculture, and algae |
| | EU Reg. 392/2013 of 29 April 2013 | • A better definition of limits and duties in control system both for operators and control bodies |
| | EU Reg. 848/2018 of 30 May 2018 | • Threshold levels of pesticides |
| NATIONAL | Legislative decree n. 220 of 17 March 1995 | • National register of organic producers<br>• Register of treatments/orchard |
| | Legislative decree n. 20 of 23 February 2018 | • Replacement of L.D. 220/1995<br>• Update of keeping records (also electronically) |

The majority of the interviewed organic food operators complain that, in any case, each control body uses its own checklist, quite often derived from a private standard, e.g., GLOBALG.A.P., very difficult to compare with others. However, overall, the transition from one control body to another becomes very hard. What was clear from the outcomes of the study was that, in most cases, the companies had no clear idea about the current legislation and its evolution. They limited themselves to the application of requests of control body and the completion of the registers, usually, paper ones. Only a few, the ones involved in mass-market, had a clear reference to sustainability and showed, for instance, records of water and fuel consumption.

With regard to the conduct of inspections in the organic food systems, a significant part of the time is used for document checks and compliance to the rules. Sometimes, the inspections are concluded with a sampling of samples and pesticide analyses. In all cases, the inspection of the control body shall be agreed. In all cases, considering that the companies chose and pay the control body, one of the weakest points is to consider it as a consultant and not as an inspector.

Regarding the public authority inspections in the organic food system, in most cases, they are unannounced inspections. This and the sampling of products are strong deterrents in preventing company misconduct. As concerns the way the public authority carries out the inspections, apart from ensuring the law requirements, overall, in the completeness of records, the inspection is strongly oriented to verify the food and supplier traceability. Traceability can be considered as the keyword in organic farm inspections.

Both in control bodies and public authority inspections, as part of their control, there are no clear references to sustainability or risk assessment.

*3.3. Requirements and Registrations Provided for Private Standards*

In the last fifteen years, the private standards and the unregulated certification schemes have assumed a central role in trade between the mass market retail (MMR) and the single or associated

companies of primary production. This is because compliance with these schemes has become almost an obligation imposed by the MMR to companies in order to be part of its supply chain, thus becoming a selection criterion [18].

The companies involved in this study think that this maneuver leads to undeniable advantages overall for the MMR, which takes advantage of extra checks carried out by the certification bodies on suppliers. However, above all, it can give more visibility and communication of the products' quality on shelves, characterizing them effectively with different brands, logos, and slogans. Moreover, another advantage could be derived by transferring costs of certification of their own-brand products to the suppliers. In fact, there is no price increase for supplies, but rather they use expert advice to implement the required certification schemes and to the extent owed to the chosen certification body for the verification of their correct application. On the other hand, they admit that such an approach has helped to become aware of the compulsory structural and documentary prescriptions foreseen for the business activity and, among these, the correct compilation of the orchard register becomes crucial.

The IFA FV GLOBALG.A.P. Standard

Currently, the IFA FV (Integrated Farm Assurance Fruit and Vegetables) GLOBALG.A.P. standard is the most common and accepted protocol by MMR in order to certify the best production techniques in farming, according to the Principles of Good Agricultural Practice (GAP). It is now globally spread and is currently applied in about 125 countries with approximately 185,000 certified companies and, in particular, with about 182,000 certified Fruit and Vegetables (FV) companies, of which a larger percentage are located in Europe and a good part operating in Italy [36]. The standard, or protocol GLOBALG.A.P., was developed in 1997 (as EUREPG.A.P.) on the initiative of retailers belonging to the Eurep working group (Euro-Retailer Produce Working Group). British retailers, working with continental European supermarkets, very soon became aware of growing consumer concerns regarding product safety, environmental impact, and the health, safety, and well-being of workers and animals. The proposed solution was to harmonize their own standards and procedures and to develop an independent and third-party certification system for Good Agricultural Practices (GAP) [37].

The aims and the purposes of the proposed protocol evolved very quickly, since the first version in 2001, already accredited ISO 65 (EN 45011) [38]. In fact, not long after, in 2003, was updated to the new Version 2, with effects from January 2004. In 2007, the transition from EUREPG.A.P. to GLOBALG.A.P. took place. The protocol review processes, implemented on average every three years, bring both new requirements and different methods to achieve compliance together with greater focuses on aspects deemed fundamental to confirm the company and its production processes to those that were the initial objectives of those who had established the protocol (Figure 1). The details and the description of each version are reported in the table in the Supplementary Material.

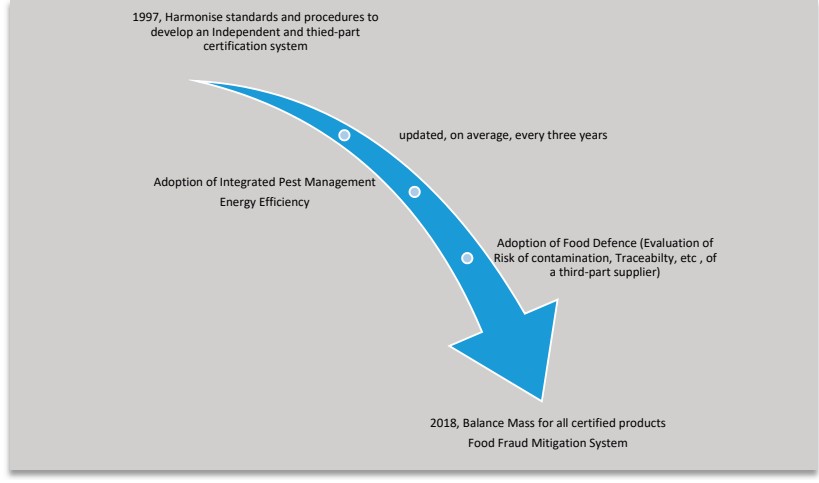

**Figure 1.** Evolution of Globalgap IFA FV Certification Scheme.

The protocol FV GLOBALG.A.P. refers exclusively to the primary agricultural production, obtained by applying the integrated farming (IF) approach, without any further processing beyond the process of selection and packaging in boxes, retinas, trays or other. The current and applied Fruit and Vegetables (FV) version is V5.2—Guideline Inspection and Justification Method, mandatory by August 2019. This new revision includes two new chapters regarding the control points/compliance criteria and the verification checklist (FV 5.9 and AF 17). The CCPs have been divided into three new major categories: Two related the conformity of the product labels GLOBALG.A.P., with respect to the food legislation of the countries in which it intends to market and the risk management of the allergens (FV 5.9.1 and 5.9.2), respectively. The third one requires a specific procedure to manage non-compliant products.

Discussion with the inspectors of the certification bodies indicate that the most important documents in the GLOBALG.A.P. scheme are:

(1)    General rules (indications on certification processes);
(2)    Control points and fulfillment criteria (requirements that must be applied by the company to achieve compliance and therefore certification);
(3)    Checklist (checklist of the application of the requirements).

The requirements of the document "Control points and fulfillment criteria" are classified in order of importance in:

-    Major (all the criteria in this section must be fulfilled);
-    Minor (at least 95% of these criteria must be fulfilled);
-    Recommendations (there are no limitations to achieve compliance).

The inspections are carried out, both as self-checks by the company and as external ones, by the qualified auditors of the chosen certification body. In any case, they have to use the appropriate checklist or the feedback list. The revision and updating process of the versions involved a continuous variation in the number and classification of the requirements, focusing the attention increasingly on: The risks related to production processes, the history of the each production site, the sustainability, and complete traceability of the products that are then finalized in a correct and timely presence of documents and above all, of records such as the orchard register, proving the correct and consistent business management.

Details of the evolution of EUREPG.A.P./GLOBALG.A.P. inspections show the introduction of Integrated Pest Management (IPM) System in Version 3.0 (July 2007), which demands food companies to adopt more sustainable management of both pesticides and infestations and to adopt practices for controlling and monitoring energy efficiency (electricity, fuel, etc.) paying particular attention to waste and introducing appropriate corrective actions. Version 4.0 (March 2011) added four new activities: Food defense, which requests a risk assessment aimed to evaluate the risk of contamination from third party (suppliers, transport, etc.), traceability and segregation, if the company produces other not certified material, they must be identified and separated the GLOBALG.A.P. number must be indicated on the label. Moreover, the integrated pest management system was adopted. Finally, version 5.1 (February 2018) adopts the balance mass for all the certified products. The Food Fraud Mitigation System aims to understand and to prevent consumers' unwise purchases. Moreover, this version introduces the necessity of the risk assessment of the production site, a sampling plan to evaluate the risk associated with pesticides (including the number of molecules, time, and sampling procedures). For the first time, a risk assessment for water irrigation, including the supplier, holding tanks, quality of used water, as well as the efficacy of irrigation system is introduced. Moreover, special attention is paid, for leaf vegetables, to the microbiological quality of water (*E. coli* and *Salmonella* spp.). The same attention is due to the use and quality of manure. The newest aspect in the checklist is the attention to allergens.

The interviews with inspectors of certification bodies highlighted that the IFA FV GLOBALG.A.P. standard is complex and constantly updated, not only to ensure food safety but also to include in

concepts of sustainability and well-being of workers. These are considered as milestones of this standard. Moreover, it was clear the difficulties and the required skills to become a GLOBALG.A.P inspector.

Analogously to the organic certification, sometimes the companies hardly recognize the certification body as an inspector rather than a consultant. The certified companies, in this case, are, on the average, much more structured and organized than the organic ones. The majority of them use management software.

The GLOBALG.A.P. checklist is considered complex. However, at the same time, comprehensive and as a useful guideline.

*3.4. The Voluntary Certification Process, Control System by Public Authorities and Applicable Sanctions.*

3.4.1. Voluntary Certification Process

Regulated product certifications, such as the organic production method.

The organic food production is a double-checking surveillance system: The public authority and a control body. After the company forwards the application to the relevant office (produced in electronic format through SIAN—National Agricultural Information System—to the SIB—Bio Information System) and electronically transmitted to the selected control body, the conversion phase to the organic method of the company starts, as foreseen by the community legislation: One year for the pastures, two years for the annual cultivations and the meadows, three years for the perennial crops. The product of the conversion phase cannot be sold as "organic". The control body has to perform, mandatorily, at least one annual inspection. This number may be increased when the company fails the checks or for the crops included in a high-risk class of the annual control plan prepared by the body. It has the right to carry out checks/inspections of the company even without prior notice and take samples of the product, both during the annual surveillance inspection and the unannounced inspection. This is aimed overall to verify the presence of pesticide residues or any other not allowed ingredient. Inspection visits include check-up of production sites and facilities and, above all, the verification of the correct identification and labeling of the productions and registrations made by the operator on the orchard register. The last one, previously issued by the control body during the initial inspection (from the raw material, cultivation, and sales sheets), from which the regular application of the provisions of the current EU regulation will be noted as well as the related national and regional implementing regulations concerning organic farming.

Unregulated product certifications, such as the IFA FV GLOBALG.A.P.

Producers can choose between two:

Option 1—Certification is requested to the chosen certification body by the individual company that shall bear the costs and the certificate, in reference to one or more company products and/or production sites.

Option 2—Certification is requested to the chosen certification body by producers' cooperatives, consortia, PO, etc. One or more companies, acts as a lead partner in the supply chain, being in charge of the: Quality manual, procedures and instructions necessary to obtain and then maintain compliance for certification.

The certification body will routinely perform an annual check which may last several days, depending on the chosen option (as described above), the number of production sites and products and whether or not the handling phase is present. All the involved phases are expected to be adequately controlled. The Option 2 provides for the annual verification as well as the management system, selecting a sample equal to the square root of the number of companies in the supply chain, by dividing the inspection into two phases: The first, so-called unannounced inspection, where the lead auditor communicates the number of companies to be sampled (half of the square root provided), without indicating the specific company subject to the inspection, the second inspection or renewal, in which the planned sampling will be completed. It is important to underline that the initial verification must be carried out compulsorily when the production process of a specific product, or groups of

similar products, and handling, is carried out. This is true also when, an already certified company, would like to add one or more new products that, in any case, must be part of the list published and updated by GLOBALG.A.P. (GLOBALG.A.P. List of Certifiable Products—CROPS (Sub-scopes: Fruit and Vegetables (FV)). The inspection checklist does not require sampling of the product, but it is mandatory that the company performs at least a pesticide analysis for each product to be certified, or, alternatively, on groups of similar products. The audit is always performed by auditors qualified by the certification body and recognized by GLOBALG.A.P. for the specific scheme [37]. The tools required to verify compliance are the two checklists published on the official GLOBALG.A.P. website: The Quality Management System Checklist (only for Options 2) and the checklist relating to the activities of agricultural holdings. The audit is performed by testing all the checkpoints in the checklists. Moreover, the auditor provides the verification of compliance with: The local and international regulations, the review of documentation, the evidence of real application of procedures and related operating instructions, the verification of recordings on a country-specific exercise book, specifically prepared by the company to assure the correct traceability of products, the application of integrated pest management and good agricultural practices, the obligations related to food safety and environmental sustainability [39]. In short, the entire corporate production process is examined so that its compliance with the requirements of the standard can be assessed and the certificate issued.

### 3.4.2. The System of Control: Authorities, the Applicable Sanctions, and the Private Standards

Conventional and/or integrated production companies: Controls and supervision on the keeping and proper compilation of the treatment or orchard registers are delegated to the health departments of the regions and autonomous provinces, to the Carabinieri Agricultural Policies Command operating with the anti-fraud units and the Central Inspectorate for the Protection of Quality and Fraud Repression of Agri-Food Products (ICQRF) [22]. The control activity is both an inspection of processes and documental type. The applicable law in this area, as mentioned above, is law no. 150 of 14 August 2012 [25] (on the implementation of Directive 2009/128/EC [24]), which specifically in art. 24, paragraph 13 introduces administrative pecuniary penalties for which, for the violation referred to the non-fulfillment of the obligations to keep the register of treatments, a pecuniary administrative sanction can be imposed from €500 up to €1500 (Table 4). Finally, it should be pointed out that the treatment register is a cross-compliance requirement to benefit from Community Agricultural Policy aids according to a specific application. All farms can be sampled and subjected to the control by AGEA (Agency for Disbursements in Agriculture). When there is a lack of respect for EU laws, this entails a percentage reduction on the premium (according to the last update Circular dated 7 August 2018 "Application of the Union and National Regulations on Conditionality").

Companies certified in organic farming: The supervision and organization of official controls for compliance with the provisions of the EU regulations and the relevant national laws concerning organic agriculture, as described in paragraph 3.2, are under the responsibility of the Italian Ministry of Agricultural, Food and Forestry Policies of Tourism (MiPAAFT). The MiPAAFT then delegates the control functions of organic farming to private control bodies authorized by it (currently there are 16 plus two of the autonomous province of Bolzano). These, in turn, are under the supervision of the Department of Central Inspectorate for the Protection of Quality and Fraud Repression of Agri-Food Products (ICQRF), by the local authorities (Regions and Autonomous Provinces) and by the MiPAAFT itself.

The control activity of organic operators by the control bodies consists of: An inspection with direct access to the sites, the control of documents, and to issue warnings and impose proportionate penalties in cases of non-compliance. The sanctions consist of an increasing level starting from the weakness, in cases where the company does not comply with the obligation to keep and compile the orchard register, up to the suspension of the certification for the single product or even of the entire company. The suspension can be also the consequence of a remark of the control body without a verifiable corrective action applied by the company. The control body, acting as a delegate of the

Ministry, in the event of serious breaches, it is obliged to inform the public authorities and, if required, do a criminal report. However, according to reports of operators, the last event is used by the control body when the companies do not pay for the controls. In the case of serious breaches, the most common consequence is the withdrawal of the organic certification.

**Table 4.** Schedule of sanctions, checks and formalities required for the keeping and/or updating of the country notebook of conventional, organic, and certified GLOBALG.A.P. companies.

| | Conventional | Organic | Certified GLOBALG.A.P. |
|---|---|---|---|
| References | Law no. 150/2012 art. 24 paragraph 13 [23] | D.M 15962/2013 [37] and Law. n. 20/2018 | Checkpoints and fulfillment criteria Current version Standard V5.1 |
| campaign notebook | Form freely processed by the company | Form from D.Lgs n. 220/95 (repealed) | Form freely processed by the company |
| Control | Departments for the Health of Regions and Autonomous Provinces, Carabinieri Agricultural Policies and ICQRF | Executed by the chosen supervisory body, by the Mipaaft and by ICQRF | Executed by the certification body chosen |
| Fertilizing | The provisions of Legislative Decree no. 150/2012 art. 24 paragraph 13 | The provisions of Legislative Decree no. 220/95 [33] | Verification checklist—control points from CB. 4 to CB. 4.2.6-Level: 7 Requirements Min. |
| Documents | Personal data relating to the company. Designation. of the culture its extension. Date of sowing, transplanting, the start of flowering and harvesting. Date of fertilization. Product and the relative quantity used. | Type of operation and date. Species and varieties on which the intervention is performed. No. of plot and the area. Used fertilizer and quantity. | Competence of the technical manager. Reference to the application site. Application date. Type of fertilizer used. Amount of fertilizer used. Application method. Operator information. |
| Sanctions | From €500 to €1500 | Please note that in the event of failure to manage the NCs, the Company may suspend until the exclusion | The total non-fulfillment of the seven minor requirements entails the Suspension of certification with consequent corrective action within 28 days. The partial non-fulfillment requires only a Warning. |
| Treatments | The provisions of Legislative Decree no. 150/2012 art. 24 paragraph 13 | The provisions of Legislative Decree no. 220/95 [33] | Verification checklist—CB 7 and 7.4.1 control points.—Level: Requirement Mag.da CB. 7.3.2 to 7.3.7—Level: Requirements Min. |
| Records | Personal data relating to the company. Designation. of the treated culture and its extension. Collection dates. Date of treatment. Product and the relative quantity used. The adversity that made treatment necessary. Name and signature of the person carrying out the treatment. | Type of operation and date. Species and varieties on which the intervention is performed. No. of plot and the area. Phytosanitary used and quantity. | The records must specify the: name of the crop and/or variety, place of application, date and time of application termination Trade name and the active ingredient of the product. Pre-collection interval. Operator name. Reason and technical authorization for the application. Amount of applied plant protection product. Application machinery used. Weather conditions. Respect of pre-harvest intervals. |
| Sanctions for default | From €. 500 to €. 1500 | Please note that in the event of failure to manage the NCs, the company may be suspended until the exclusion | The non-conformity of even one of the two Mag. Requirements entail the Suspension of certification with consequent corrective action within 28 days. The total non-fulfillment of the 6 Minor requirements entails the Suspension of certification with consequent corrective action within 28 days. The partial non-fulfillment requires only a warning. |

The obligations under the organic farming are reported in the Regulation (EU) n. 392/2013 of the Commission of 29 April 2013 [30] (which defines the non-conformities and the corresponding sanctions applicable by the control bodies to the organic operators). Namely, in art. 1 par. 6 complements EC Reg. 889/2008 [29] with article 92 quinquies "List of measures in case of irregularities and infringements" that the control bodies must apply to the operators.

In Italy, these laws have been applied by Ministerial Decree No 15962 of 20 December 2013 [40] (which categorizes the non-conformities, irregularities, and infractions, involving the warning, the suppression of the organic designation, the suspension of certification and, lastly, the exclusion). A special mention must be reserved for the Ministerial Decree No 18096 of 26 September 2014 [41], which defines the timing and the management of the measures to be adopted by the control bodies

as concerns the organic operators in the implementation of Ministerial Decree 15962 of 20 December 2013. The very newest Legislative Decree, no. 20 of 23 February 2018 [36], aimed to a harmonization of the legislation on controls of organic agricultural production and agro-food. Namely, this decree provides for an intensification of the supervisory functions by the Ministry and the ICQRF. However, overall, it introduces the concept of "shared responsibility" of the involved parties, defining pecuniary sanctions, sometimes very onerous, both on the controlled organic companies and the respective control body. This happens when the control body did not explicitly refer to penalties for failure to comply with the requirements for keeping the treatment register.

GLOBALG.A.P. IFA FV Standard: This is a voluntary non-regulated product certification scheme, legally binding only on the basis of a contractual agreement between the company and the certification body.

The control for compliance with the requirements is the responsibility of the chosen certification body, without any intervention by the public authority, except in cases in which the detection of non-conformities deemed to be serious for public safety. As previously stated, compliance to obtain or maintain the renewal of certification is subject to compliance with 100% of all applicable control points with a major requirement and at least 95% of all applicable control points with a lower requirement. The detection of non-conformity during the audits performed by the certification body, classified as minor or recommendations, involves the notice of a warning. Then, the company is asked to manage it with appropriate corrective actions to be verified during the following audit.

Initial verification: The presence of major NC or a high number of minors NC, so that the total applicable complaint is lower than 95%, obliges the company to manage them with adequate corrective actions within 28 days. Three months without providing any document and/or any result, the certification body needs to plan a new complete visit before issuing the certificate.

Periodic inspections: In this case, both the presence of major NC and a high number of minors NC such that the total applicable complaint is lower than 95%, obliges the company to manage them with adequate corrective actions within 28 days. If the NCs are not managed within this deadline, the warning is changed into the suspension of company certificate. The suspension, as a consequence, requires the removal of certification references from tax documents, labels, and product packaging. Moreover, it prescribes within a maximum of 12 months, the resolution of the NC has been positively addressed. After that, the certification body will directly propose the cancellation of the contract. The company that has been sanctioned with a cancellation cannot request new GLOBALG.A.P. certification, nor to the certification body that has sanctioned them or to other bodies, not before 12 months from the cancellation date [42].

## 4. Conclusions

The diffusion of the certification schemes of food products on a voluntary basis today is a consolidated reality, representing a sector in constant growth in the agri-food panorama. The building block of this success is certainly the high level of guarantee in terms of food safety and quality of certified product delivered to the final consumer, together with even a higher perception of quality and a commitment to improving sustainability. This study highlighted how the regulatory framework of low impact food production is constantly evolving and gaining complexity. It has a deep impact on the practical application, both for food operators and inspectors, and, ultimately, on the very concept of sustainability. The main outputs are:

- It is essential to document the agriculture commodities and final product traceability. Moreover, it is important to correctly execute the mandatory registrations required by specific regulations. For this purpose, both the orchard register and the company register play a crucial role.
- Comparing the procedures of control of organic farming with those of the voluntary non-regulated Voluntary Certification Protocol IFA FV, it is possible to highlight differences in the constraints of information to be recorded which then lead to cases of partial or total non-compliance with different degrees of sanctions.

- The different standards are not equivalent also in the consequences of misconduct: The consequences can go from the withdrawal of certification up to criminal charges.

- It emerges that farms certified with voluntary non-regulated certification systems, such as the IFA FV GLOBALG.A.P., to obtain or confirm the certification are obliged to take into account the highest number of aspects, reported in a more complete register than the organic ones. In fact, it contains much more important information than that adopted by the other system, showing a readier attitude regarding the needs required by the market and the restrictions and innovations imposed by the related sector legislation. Moreover, this is also supported by a continuous-time planned process of revision and updating of the applicable versions of the standard, which is clearly much more dynamic with respect to the voluntary legislation that regulates the organic production method in dealing comprehensively with the production phase of food products.

- From this study, it clearly emerges that the protocol GLOBALG.A.P. takes into account many more aspects than the organic protocol. The environmental impact of the food production, the safety aspects of food products, as well as the health, ethics, and safety aspects of workers, are largely considered and inspected in the GLOBALG.A.P., while organic system, despite the IFOAM suggestions and indications [32], is considered partially. This means that, from a practical point of view, the organic product can be considered "clean and safe", but not more environmentally friendly than the GLOBALG.A.P. products.

Finally, the authors do think that an assessment study, based on FAO SAFE guidelines [6], in the same way as what has been done in assessing the sustainability of different urban food system, would help to understand the degree of sustainability both of organic and GLOBALGAP food production systems.

**Supplementary Materials:** The following are available online at http://www.mdpi.com/2071-1050/12/2/606/s1. Table S1. Control points provided for by the different versions of the Eurepgap/GLOBALG.A.P. IFA FV protocol.

**Author Contributions:** S.S. has made substantial contributions to the acquisition and analysis of data; E.A. has made substantial contributions to the interpretation of data and draft of the work; R.P. has made substantial contributions to the interpretation of data and draft of the work; B.F. has made substantial contributions to the conception, design, draft and revise of the work. All authors have read and agreed to the published version of the manuscript.

**Funding:** This research received no external funding.

**Conflicts of Interest:** The authors declare no conflict of interest.

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
