# Peer review of "Public and Private Standards in Crop Production: Their Role in Ensuring Safety and Sustainability"

_sustainability, doi:10.3390/su12020606_

Round 1
Reviewer 1 Report
L'argomento dell'articolo è molto interessante.
ASTRATTO
Nelle righe 12-15 ci sono ripetizioni di concetti.
MATERIALI E METODI
Il paragrafo non è molto approfondito. È necessario illustrare meglio il questionario utilizzato e la fase di raccolta dei dati, durata 24 mesi
RISULTATI E DISCUSSIONI
Le tabelle 2 e 3 sono una semplice descrizione della diversa versione? Se sì, si suggerisce di spostarlo in allegato perché, a mio avviso, rischia di gravare sulla lettura Una tabella riassuntiva dovrebbe essere inserita per chiarire meglio i risultati. La notevole quantità di recenti indicazioni legislative e, in alcuni punti, non rischia di non comprendere appieno i risultati della ricerca. I risultati presentati provengono da interviste condotte nel corso dei 24 mesi? Non è ben chiarito.Author Response
1) The abstract has been modified as follow:
- Ln 11, the repetition has been deleted;
- Lns 14-15, the sentence “This study shows that the protocol GLOBALG.A.P. takes into accounts much more aspects than the organic protocol.” Has been deleted;
2) Materials and Methods has been significantly improved
- The table with the questionnaire used in the interviews has been added.
3) Results and Discussion
- Table 1 has been deleted
- Table 2 moved to supplementary material
- New simplified tables and a new Figure have been added
- The discussion of data, where it was possible, has been simplified
4) Literature: has been update with the addition of two new reference.
Reviewer 2 Report
The aims of the study are therefore to: provide an overview of the binding regulation of the sector and of the operating procedures concerning the registration of the cultivation activities of primary products by the agricultural operators; to verify the different annotation modalities envisaged between the organic and the conventional production methods with reference to the obligations imposed by the respective regulations and then comparing them with the corresponding registrations foreseen by private certification schemes, such as the IFA FV GLOBALG.A.P.. Moreover, the Control and Supervisory Systems operating by public or private authorities and the sanctioning systems was highlighted. From this study clearly emerges that the protocol GLOBALG.A.P. takes into accounts much more aspects than the organic protocol. The Environmental Impact of the food production, the Safety aspects of food products, as well as the Health, Ethics and Safety aspects of workers, are largely considered and inspected in the GLOBALG.A.P., while Organic system, although the IFOAM suggestions and indications, considers partially. This means that, from a practical point of view, the organic product can be considered “Clean and Safe”, but not more environmentally friendly than the GLOBALG.A.P. products.
The paper is quite interesting but I have some remarks to improve the paper understanding.
The paragraphs from 3.1 to 3.3 could shift to section 2. Materials and Methods
The tab 2. Control points provided for by the different versions of the Eurepgap / GLOBALG.A.P. IFA 333 FV protocol the authors have to improve the editing
It would be interesting if the authors could add references to the following works that, based on the food processing buildings sustainability , namely:
https://www.tandfonline.com/doi/abs/10.1080/2093761X.2012.759888
https://www.mdpi.com/2071-1050/11/17/4601
There are some english mistake and multiple grammatical errors and spelling mistakes in paper (e.g. - CERTIFICATION CERTIFICATE on line 11). Language editing is a must for revision
Author Response
1) The abstract has been modified as follow:
- Ln 11, the repetition has been deleted;
- Lns 14-15 the sentence “This study shows that the protocol GLOBALG.A.P. takes into accounts much more aspects than the organic protocol.” Has been deleted;
2) Materials and Methods has been significantly improved
- The table with the questionnaire used in the interviews has been added.
3) Results and Discussion
- Table 1 has been deleted
- Table 2 moved to supplementary material
- New simplified tables and a new Figure have been added
- The discussion of data, where it was possible, has been simplified
4) Literature: has been update with the addition of two new reference.
5) We think that paragraphs 3.1-3.3 should stay in chapter 3 as they are the outcomes both of the study of food legislation and the interviews with food companies and, overall, with the public Authority
Reviewer 3 Report
The authors of the research paper "Public and private standards in crop production: their role in ensuring safety and sustainability" address a topic of current issues in the context of the 2030 UN Agenda.
The bibliographic tops and the specialty literature are properly used, especially in the context of the work which is more oriented towards concepts, regulations and standards and less on the classical documentation specific to a research paper (this is a limitation).
The research methodology is based on methods and tools appropriate to the given topic, respectively by using the structured and target group questionnaire involved in the activity of certifying organic products (relevant and detailed information on crop management practices, pesticide use and handling). At the same time, given the theoretical nature of the paper, we propose to the authors of the research and an analysis of the risks and limitations of the risks of applying public and private standards in crop production, as well as their role in ensuring safety and sustainability.
The results of the research are adequately presented considering the topic of the paper, respectively the comparative analysis of the regulations and / or standards for the organic production method are presented.
The conclusions are adequately presented from the point of view of the investigated aspects, as well as the usefulness of these standards at the farm level, but we suggest to the research authors a clear presentation of their own scientific contributions in the paper.
Therefore, with these additions / revisions made by the authors, we propose, after the revision, the acceptance for publication of the work proposed by the authors.
Author Response
In the Introduction, we added two new references concerning the assessment of sustainability. We added some new comments concerning the results of interviews and, overall, in the conclusion section, we have tried to highlight the findings of this research.We appreciated very much your suggestion about a risk assessment. An LCA assessment is already in progress (considering fuel, water, electricity, fertilizers, etc.., consumptions). Considering your suggestion, we found a very useful method (references 5 and 6) aimed at a broader sustainability assessment. But, at the moment, we did not have all the requested data.
Best regards
Biagio Fallico
Round 2
Reviewer 1 Report
Gli autori hanno apportato le modifiche suggerite e migliorato la leggibilità dell'articolo
Author Response
Thank You
Reviewer 3 Report
After the review, we propose for acceptance and publication the paper and congratulate the research team for the topic of the paper.Author Response
Dear Reviewer,
Please, enclosed the revised manuscript.
Best regards
Biagio Fallico